# Geometrically Consistent Generalizable Splatting

## Abstract

Gaussian splatting has emerged as the preferred 3D scene representation due to its incredible speed and accuracy in novel view generation. Various attempts have thus been made to adapt multi-view structure prediction networks to directly predict per-pixel 3D Gaussians from images. However, most work has focused on enhancing self-supervised depth prediction networks to estimate additional parameters for 3D Gaussians – orientation, scale, opacity, and appearance. We show that optimizing a view-synthesis loss alone is insufficient to recover geometrically meaningful splats in this simple manner. We systematically analyse and address the inherent ambiguities in learning 3D Gaussian splats with self-supervision to learn pose-free generalisable splatting. Our approach achieves state-of-the-art performance in (i) geometrically consistent reconstructions, (ii) relative pose estimation between images, and (iii) novel-view synthesis on the RealEstate10K and ACID datasets. We also showcase zero-shot capabilities of the proposed generalizable splatting on ScanNet, where our method substantially outperforms the prior art in recovering geometry and estimating relative pose.

## 1 Introduction

3D Gaussian splatting (3DGS) [22] has recently revolutionized 3D structure and appearance modeling from multi-view images. Departing from traditional depth or point cloud representations of the scene structure, 3D Gaussians implicitly model surface reflections and environment lighting to encode view-dependent scene appearance. They are memory-efficient compared to explicit volumetric alternatives, and they facilitate rendering of the scene from arbitrary viewpoints in a fraction of a second. Due to these capabilities, 3D Gaussians have become a prevalent choice for scene representation.

Learning-based structure estimation methods, such as single- or two-view depth predictors, are increasingly being adapted to directly predict 3D Gaussians using feedforward neural networks. Various laudable attempts have been made recently in training neural networks to predict 3D Gaussians directly from images, achieving photorealistic results without per-scene optimization [2, 3, 43, 47, 35, 54, 51]. These methods are commonly referred to as *generalizable Gaussian splatting*. Most generalizable Gaussian splatting methods adapt well-studied one- or multi-view structure prediction networks [46, 42, 25] to estimate locations of the 3D Gaussians. These networks typically use image encoders that take in one or multiple images followed by decoders that predict Gaussian means, in the form of per-pixel depth [46] or 3D point locations [42, 25] for each input view. Nearly all generalizable Gaussian splatting methods append additional decoders to depth or point-cloud estimation architectures to predict Gaussian properties such as orientation, scale, opacity and view-dependent color – typically without much foresight. These networks are usually trained by minimizing view-synthesis loss on a few target views, closely following existing self-supervised depth estimators – though they differ in image formation due to the underlying change in scene representation.

This prevalent setup overlooks several key issues inherited from the underlying 3DGS optimization:

Submitted to 39th Conference on Neural Information Processing Systems (NeurIPS 2025). Do not distribute.

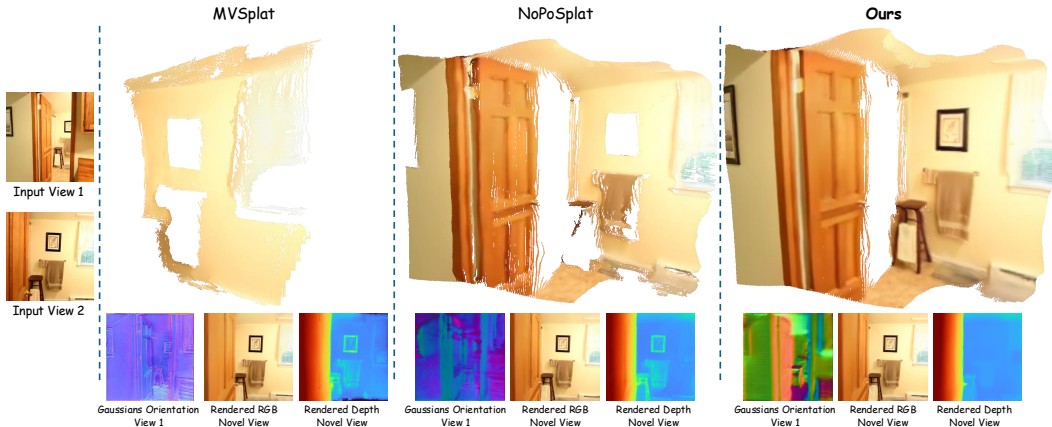

Figure 1: **Qualitative comparison of mesh reconstructions from two input views.** We compare the baseline methods MVSplat [3] and NoPoSplat [51] with our approach on RE10K dataset [57]. The *top row* displays the textured meshes reconstructed by fusing *virtual* depth maps via TSDF; the *bottom row* visualizes Gaussian-surface normals for the first input view, and the RGB/depth rendered from a novel (virtual) viewpoint. Inaccuracy in rendered depth and normals is evident for both baselines. These inconsistent depths, when fused, create several holes in the mesh reconstruction. Our method recovers accurate geometry of fine details such as the towel, wall painting, and stool.

- 3D Gaussians are grossly overparameterized compared to depth maps or point clouds. Successful estimation of 3D Gaussians typically requires a large number of densely sampled viewpoints. Few-shot 3DGS is an active field of research and often relies on regularization priors [59, 4], which are largely ignored when training generalizable splatting networks.

- Unlike per-pixel depths or 3D point locations – which are uniquely defined (up to scale) – multiple 3D Gaussian configurations can produce equally valid renderings. This inherent ambiguity makes training difficult, even when depth data is available for supervision.

- Successful per-scene Gaussian splatting methods typically rely on multiple non-differentiable heuristics (*i.e.*, splitting, and duplication of Gaussians). However, existing generalizable methods are trained purely via view-synthesis gradient loss and neglect these heuristics. i.e. they assume that all Gaussians remain perpetually alive during training.

As a result, existing generalizable Gaussian splatting methods often converge to *geometrically degenerate Gaussians*. While the predicted locations (means) remain relatively stable —- benefiting from well-established single- or multi-view depth estimators – other parameters (opacity, orientation, scale) are prone to collapse. As shown in Figure 2, existing generalizable approaches struggle to learn meaningful opacities, orientations, or scales when trained with view-synthesis loss alone in both pose-aware and pose-free settings. In particular, we observe implausible Gaussian orientations (in the form of normals) as well as unjustified elongation of the 3D Gaussians (scales).

We show that these artifacts are due to the inherent over-parametrization of geometry in the form of splats, which require structural consistency priors to make the self-supervised learning viable. By introducing such priors, our proposed method produces Gaussians that exhibit consistent and physically plausible geometric patterns. As shown in Figure 2, our proposed method produces accurate surface normals directly from the predicted Gaussian orientations. The resulting Gaussians – parameterized as 2D disks in 3D space – are elongated along geometric discontinuities and remain robust to image textures. The two approaches we selected for visualizing predicted 3D Gaussians broadly represent distinct underlying representations for encoding Gaussian means: (i) per-pixel depth maps [3], and (ii) per-pixel 3D locations aligned to a common reference frame [51]. Despite this difference, to our knowledge, all existing self-supervised generalizable splatting methods suffer from similar limitations – stemming from their reliance on the representations and loss functions introduced in [3, 51].

In this work, we aim to systematically define the ideal configuration of a geometrically consistent Gaussian and propose appropriate priors to assist generalized Gaussian splatting. To that end, we opt to build upon the recently proposed NoPoSplat framework [51] as our baseline. We choose

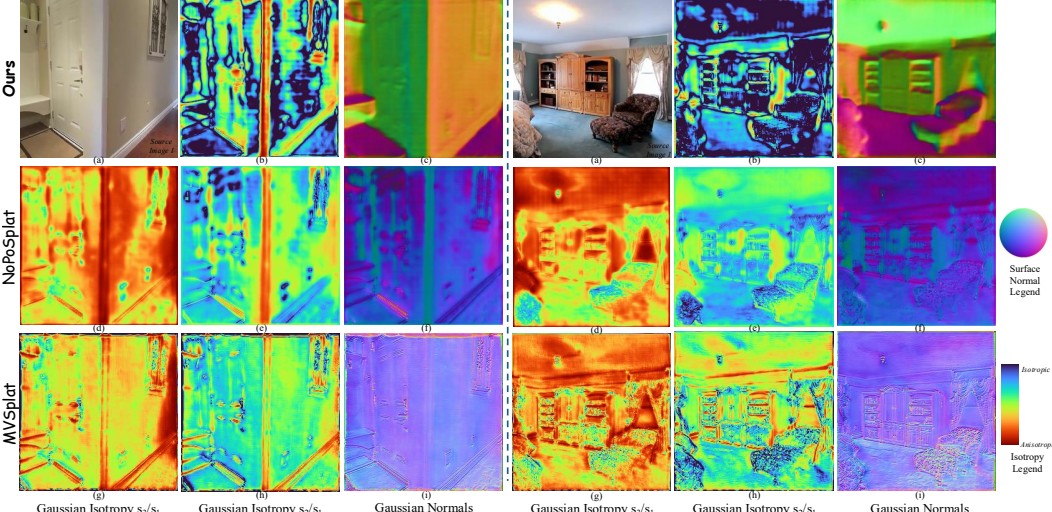

Figure 2: **Qualitative comparison of predicted Gaussian parameters**. Each Gaussian has scales $s_1 > s_2 > s_3$, where $s_3$ quantifies the uncertainty in localization of the surface the Gaussian belongs to with surface orientation defined by its normal [18]. **Row 1 (ours)** shows: (a) the source image to which Gaussians are aligned, (b) skewness of the estimated Gaussians within their own defining plane, and (c) predicted Gaussian orientations visualized as surface-normal maps. **Rows 2 and 3** show results for NoPoSplat and MVSplat, respectively: (d/g) Gaussians' elongation perpendicular to the dominant plane defined by it, (e/h) Gaussians' skewness within the dominant plane, and (f/i) normals to the dominant plane. Existing methods yield Gaussian orientations without clear geometric meaning: MVSplat Gaussians (i) align mostly fronto-parallel to the source image plane, and NoPoSplat Gaussians orientations (f) strongly depend on texture, spanning a few dominant directions inconsistent with scene geometry. Our method produces plausible, near-Manhattan structured surface orientations. Baseline Gaussians exhibit significant elongation perpendicular to their dominant surfaces (visible as non-red colors in d/g). Notably, our Gaussians remain relatively circular (blue color in b) on planar, textureless surfaces and become skewed ellipses (red color in b) near sharp geometric edges such as shelves or wall corners.

NoPoSplat not only for its state-of-the-art performance across relative camera pose estimation and view-synthesis but also for its self-supervised formulation, which does not require groundtruth depth maps. Additionally, utilizing DUSt3R [42] framework, the approach is one of the few to provide generalizable splatting from a pair of images without requiring the relative pose of these images.

We address ambiguities in learning over-parametrized 3D Gaussians by adding suitable regularization terms to the traditionally used view-synthesis loss. Our main observations are:

- Defining the ideal 3D Gaussian orientations to be dominant normals of the scene surfaces (as in [18]) helps resolve structural ambiguities to enable learning of the Gaussian orientations.

- Ensuring that the 3D Gaussians are pixel-aligned is extremely important with self-supervision. Particularly pose-free methods that use PnP for relative pose estimation require pixel alignment to ensure accurate camera pose and geometry prediction.

- Standard priors used to enforce consistency in rendered depth and normal maps [18] are not easily deployable for joint learning of pose and structure. Instead, we enforce consistency between Gaussian orientations and means by leveraging the local image neighborhood of pixel-aligned Gaussians. This promotes stable training in generalizable splatting networks.

We build upon NoPoSplat [51], integrating geometric consistency priors, and trained our network on the RealEstate10K (RE10K) dataset [57]. Our method outperforms prior work in novel-view synthesis and, importantly, produces plausible scene geometry that enables direct depth rendering from arbitrary viewpoints – something current methods cannot achieve. These consistent virtual depths can be fused using Truncated Signed Distance Function (TSDF) [55] and the reconstructed meshes are visualized for comparison with prior art in Figure 1. Our approach also establishes a new

state-of-the-art in relative pose estimation from image pairs, surpassing methods with task-specific training such as RoMa [6], geometry-supervised approaches such as DUSt3R [42, 25], and pose-free generalizable splatting method [51] – despite using less data and weaker supervision in some cases. Our approach achieves state-of-the-art zero-shot 3D reconstruction on ScanNet [5], outperforming all existing pose-free and pose-required generalizable splatting methods.

To the best of our knowledge, this is the first work in the domain of generalizable Gaussian splatting which systematically analyze and evaluate the veracity and geometric meaningfulness of predicted Gaussian orientations and elongations. We address the shortcomings of existing approaches in training generalizable splatting networks to produce Gaussians that enable accurate depth rendering from virtual views. We believe that the presented analysis lays the groundwork for future research on training neural networks to predict Gaussions form images, in both depth-supervised and self-supervised setups.

## 2    Related Work

Owing to the state-of-the-art real-time view synthesis performance of 3D Gaussian splitting [22], significant effort has been put into improving 3DGS for scenarios such as few-view reconstruction [59, 4, 26, 16, 45, 39], dynamically moving objects [44, 49, 48, 29], surface extraction [15, 18], and incorporating object semantics into 3D reconstructions [27]. Real-time simultaneous localization and mapping approaches have also adapted Gaussian splats as an inherent scene representation [31, 21]. Additionally, Gaussian splats have been used for generating geometrically consistent images and video sequences [41, 44].

The deep learning revolution of the last decade has significantly influenced geometric inference from one or more images. Earlier works focused on training neural networks to map a single image to depth map obtained from range sensors [8, 23, 9, 24, 33]. Multi-view extensions for these supervised learning algorithms are well explored as well [1, 19, 50, 14, 40, 38]. More recently, methods have explored reconstructing registered sets of per-pixel point clouds from multiple images, providing state-of-the-art relative pose and scene structure [42, 25].

Additionally, it has been demonstrated that these feed-forward geometry prediction networks can be trained without depth sensors in a self-supervised manner by minimizing view synthesis losses [11, 12, 56, 53, 13]. Structure prediction from single or few images has also been utilized as an optimization-free building block in high-fidelity tracking and mapping systems [58, 60]. Generalizable Gaussian Splatting methods have evolved recently to learn neural networks that predict 3D Gaussians explaining a scene directly from a few images. We broadly categorize these methods into following two categories:

**Pose-Dependent Generalizable 3DGS**: Several works assume input images come with known or pre-computed poses (e.g., via SfM) and focus on designing architectures to infer 3D Gaussians from these posed views [2, 3, 32, 43, 47, 10, 37, 54]. A prominent example is pixelSplat [2], which introduced a two-view feed-forward network that utilizes epipolar cross attention transformer architecture to fuse multi-view information and predict per-pixel depth distribution for input images. This distributions are sampled to create a set of 3D Gaussian centers along the viewing rays. MVSplat [3] uses cost volume based fusion of multi-view information, adapting the Unimatch [46] architecture to regress for depth instead. Both methods use additional decoder heads to estimate rest of the 3D Gaussian parameters.

**Pose-Free Generalizable 3DGS**: An emerging frontier involves dispensing with known camera poses—allowing the network to infer scene geometry and camera registration jointly from images alone [35, 20, 51]. Early efforts in this direction often build upon learned stereo matching. For example, [35] tackles uncalibrated stereo pairs by extending a foundation model (MASt3R [25]) that predicts dense point clouds from two images. It then outputs 3D Gaussians directly in a canonical frame, augmenting each point in the MASt3R reconstruction with color and covariance attributes. This process is supervised using the geometry of the 3D point cloud and followed by a novel-view synthesis stage to fine-tune appearance. NoPoSplat [51] adopts a more self-supervised, multi-view approach by anchoring one view's coordinate system as canonical and training a network to predict all Gaussians directly in that space, using only a photometric loss for training.

To the best of our knowledge, all aforementioned generalizable splitting methods struggle to learn geometrically faithful orientations and scales for 3D Gaussians. The proposed approach alleviates this issue from generalizable splatting using appropriate geometric priors.

## 3 Method

In this section, we present our generalizable Gaussian splatting framework and loss functions we propose to address the ill-posed nature of self-supervised learning in predicting geometrically consistent Gaussians. For the architectural details, we refer the reader to supplementary material.

**Problem Definition** Assuming that we are given a set of sparse images $\mathcal{I} = \{\boldsymbol{I}_t \in \mathbb{R}^{H \times W \times 3}\}_{t=1}^{T}$ (which is also known as context images in [2, 35, 3, 51, 47, 20]), each with known camera intrinsics that form the set $\mathcal{K} = \{\boldsymbol{K}_t \in \mathbb{R}^{3 \times 3}\}_{t=1}^{T}$ capturing a *rigid* scene, our aim is to learn a feedforward neural network $f_{\Theta}$ that maps these images and intrinsics $(\mathcal{I}, \mathcal{K})$ to a set of *pixel-aligned* Gaussians as

$$f_{\Theta}(\mathcal{I}, \mathcal{K}) = \left\{ \mathcal{G}_t^j := \left( \boldsymbol{\mu}_t^j, \alpha_t^j, \boldsymbol{q}_t^j, \boldsymbol{s}_t^j, \boldsymbol{c}_t^j \right) \right\}_{t=1:T}^{j=1:H \times W}, \tag{1}$$

where $\mathcal{G}_t^j$ is the 2D Gaussian defined in the 3D space corresponding to a pixel $j$ in image $t$. Each $\mathcal{G}_t^j$ is characterized by its *center* $\boldsymbol{\mu} \in \mathbb{R}^3$; *orientation* represented by a unit quaternion vector $\boldsymbol{q} \in \mathbb{R}^4$; two *scale* parameters $\boldsymbol{s} \in \mathbb{R}^2$ defining the elongation of the 3D Gaussians; *opacity* $\alpha \in \mathbb{R}$; and *color* encoded as spherical harmonics $\boldsymbol{c} \in \mathbb{R}^d$. In this work, we advocate the use of 2D Gaussians [18] to represent the scene instead of the standard 3D Gaussians adopted by prevalent generalizable Gaussian splatting frameworks [2, 3, 47, 51]. Following [18], we assume that estimated Gaussians are aligned with the scene surface and its elongation perpendicular to the local surface normal is zero. We show through extensive evaluations how this choice helps generalizable Gaussian splatting in Section 4.

Note that both Gaussian centers $\boldsymbol{\mu}_t^j$ and orientations $\boldsymbol{q}_t^j$ are defined in the image coordinates of the *first* image $\boldsymbol{I}_1$. Given these $M \times N \times T$ Gaussians predictions, we render *novel views* of the scene $\{\hat{\boldsymbol{I}}_f \in \mathbb{R}^{H \times W \times 3}\}_{f=1}^{F} \subset \mathcal{I}$ from $F$ different viewpoints defined by its projection matrix $P_f = (\mathbf{R}_f, \mathbf{T}_f) \in \mathrm{SE}(3)$ to be matched with its observed images $\mathbf{I}_f$s during training.

We propose to minimize the view synthesis loss [51, 3, 2] from the predicted Gaussians as

$$\mathcal{L}_{synthesis} = \sum_{f=1}^{F} \mathcal{L}_{rgb}(\mathbf{I}_f, \hat{\mathbf{I}}_f) + \mathcal{L}_{lpips}(\mathbf{I}_f, \hat{\mathbf{I}}_f), \tag{2}$$

where $\hat{\mathbf{I}}_f(u, v)$ is the color corresponding to a pixel $(u, v)$ in image $\mathbf{I}_f$ rendered by blending $K$ ordered *projected* Gaussians $\mathcal{G}'$ using the 2DGS rasterizer as

$$\hat{\mathbf{I}}_f(u, v) = \sum_{k=1}^{K} \boldsymbol{c}_k \alpha_k \mathcal{G}'(u, v) \Pi_{j=1}^{k-1}(1 - \alpha_j \mathcal{G}'(u, v)), \tag{3}$$

Note that $\mathcal{G}'$ is the projection of the Gaussians $\mathcal{G}$ onto the 2D image plane of the image $\mathbf{I}_f$, see supplementary material for more details.

As shown in Section 4, solely relying on view synthesis loss is proven to be insufficient for learning geometrically meaningful Gaussians. In this work, we propose to minimize two additional regularization losses: (i) a depth-surface normal consistency term $\mathcal{L}_{orient}$ to align the orientations of the Gaussians with the rendered depth; (ii) a grid alignment loss $\mathcal{L}_{align}$ to ensure that the estimated Gaussians are aligned with the pixels of the provided images. Combining these two regularization with the view synthesis loss, we define our training objective function $\mathcal{L}_{total}$ as

$$\mathcal{L}_{total} = \mathcal{L}_{synthesis} + \lambda_o \mathcal{L}_{orient} + \lambda_a \mathcal{L}_{align}, \tag{4}$$

where $\lambda_o$ and $\lambda_a$ are weighting factors balancing the influence of each regularization. We discuss the motivation, formulation and impact of the regularization term in the following sections.

### 3.1 Learning Gaussian's Orientations.

Recall that existing pose-free and pose-aware generalized Gaussian splatting approaches struggle to learn meaningful Gaussian's orientations, see Figure 2. To provide geometric meaning for the

orientations of the Gaussians, we propose to align them with the dominant surface normals of the scene they belong to. To that end, we follow the setup in [18] to estimate only *two non-zero Gaussian scales* $(s_t^{j,1}, s_t^{j,2})$ and set the third $s_t^{j,3}$ (along the normal) to zero, so each Gaussian is "flat" in one direction. The resulting rank-deficient Gaussian covariance matrix $\Sigma_t^j$ is defined as

$$\Sigma_t^j = \mathbf{R}(\boldsymbol{q}_t^j) \, \text{diag}([s_t^{j,1}, s_t^{j,2}, 0]^T) \, (\mathbf{R}(\boldsymbol{q}_t^j))^T, \text{where } \mathbf{R}(\boldsymbol{q}_t^j) \in \text{SO}(3) \tag{5}$$

whose null space – the zero-eigenvalue direction – encodes the Gaussian's surface normal that is predicted by the network $\boldsymbol{N}_t^j$.

Swapping 3D Gaussians to 2D surface elements reduces the over-parameterization to an extent; however, the view synthesis loss of eq. (3) does not provide a sufficiently strong supervision signal for learning orientations. The most natural way to supervise Gaussian orientation is to use surface normal regularization prior from [18], where the authors propose to enforce consistency between rendered normal and rendered depth maps while doing splatting. Naively deploying such regularization to train our model does not work well. In the supplementary material, we discuss our observations and provide remedies for successfully adapting such regularization.

Instead, we propose to use a simple yet effective alternative to supervise the predicted Gaussian orientation $\boldsymbol{q}_t^j$. Leveraging the assumption that each Gaussian $\mathcal{G}_t^j$ is aligned with the 2D image pixel $j = (u, v)$ in image $t$, we define the *local surface normal* for $\mathcal{G}_t^j$ using the 3D positions of its neighboring pixels as $\hat{\boldsymbol{N}}_t^j$. We enforce these estimated local surface normals to be consistent with the predicted normal $\boldsymbol{N}_t^j$ (null space of the convenience matrix $\Sigma_t^j$) by minimizing the following loss

$$\mathcal{L}_{orient} = \frac{1}{T(H-2)(W-2)} \sum_{t=1}^{T} \sum_{u=2}^{W-1} \sum_{v=2}^{H-1} \|1- < \boldsymbol{N}_t^j, \hat{\boldsymbol{N}}_t^j > \|_\rho, \tag{6}$$

$$\hat{\boldsymbol{N}}_t^j = \|(\boldsymbol{\mu}_t^{(u+1,v)} - \boldsymbol{\mu}_t^{(u-1,v)}) \times (\boldsymbol{\mu}_t^{(u,v+1)} - \boldsymbol{\mu}_t^{(u,v-1)})\|_*, \tag{7}$$

where, $< ., . >$ and $, . \times .$ are dot and cross-product or two vectors, $\|.\|_*$ represents vector normalization and $\|.\|_\rho$ is Huber loss (implemented as SmoothL1Loss in Pytorch).

Note that, unlike the loss used in 2D Gaussian splatting (2DGS) [18], $\mathcal{L}_{orient}$ does not involve any rasterization, providing a direct supervision for orientation given the Gaussian means. In fact, the proposed loss mimics the standard loss used for supervised learning of surface normals [7], where the ground truth normals are estimated from the depths using eq. (7). This simple loss in our experiments outperforms alternatives and can be used for depth supervised training of generalizable splats as well.

## 3.2 Pixel-aligned Gaussians

Although the first generalizable splatting approach [2] worked in a pose-aware setup and adapted two-view depth prediction network, they by construction constrains every Gaussians to lie on its corresponding viewing ray. Pose-free variants [51] drop the camera pose assumption by directly estimating the Gaussian's locations in the canonical space using a DPT decoder. While this removes the need to warp Gaussians with known cameras, the parametrization rendered the structure estimation problem ill-posed especially under self-supervised regime. Specifically, in contrast to depth-supervised frameworks like DUSt3R [42], which learns an implicit structural prior by enforcing the reconstructed 3D point cloud to project onto the regular image grid, the view synthesis loss in eq. (3) does not offer such constraint. Gaussians can therefore move freely into geometrically degenerate configurations, hampering both structure and relative pose estimation.

Therefore, we explicitly align each Gaussians to with its pixel's viewing ray. Specifically, for each pixel $(u, v)$ in frame $t$, the Gaussian's centers $\boldsymbol{\mu}_t^{(u,v)}$ must be projected to that pixel location with known camera extrinsics $(\mathbf{R}, \mathbf{T})$ and intrinsic matrix $\mathbf{K}$. We enforce this with the alignment loss as

$$\mathcal{L}_{align} = \frac{1}{\sum_{t,u,v} \mathcal{M}_t^{(u,v)}} \sum_{t=1}^{T} \sum_{u=1}^{W} \sum_{v=1}^{H} \mathcal{M}_t^{(u,v)} \|[u, v]^T - \Pi(\mathbf{K}_t[\mathbf{R}_t|\mathbf{T}_t]\boldsymbol{\mu}_t^{(u,v)})\|_2^2; \tag{8}$$

where $\mathcal{M}_t^{u,v}$ is 1 if the Gaussian projects inside the image and have positive depths (otherwise 0) and $\Pi([X, Y, Z]^T) = [X/Z, Y/Z]^T$ is the perspective projection function. We demonstrate in Section 4 that the proposed loss plays a crucial role in PnP-based relative pose estimation (Table 1) as well as accurate structure estimation (Table 3).

Table 1: **Pose estimation (AUC) at multiple error thresholds on RE10K [57] (in-domain) and on ScanNet-V1 [5] and ACID [30] (cross-domain).** The overall best results are shown in **bold**, and the best result-whether with or without photometric optimization-is underlined in each section. Methods marked with † are trained on additional data (e.g., ScanNet, ACID), and those marked with ‡ use extra supervision (e.g., ground-truth depth).

| | | RE10K | | | ScanNet-V1 | | | ACID | | |
|---|---|---|---|---|---|---|---|---|---|---|
| | Method | 5° ↑ | 10° ↑ | 20° ↑ | 5° ↑ | 10° ↑ | 20° ↑ | 5° ↑ | 10° ↑ | 20° ↑ |
| | CoPoNeRF† | 0.161 | 0.362 | 0.575 | - | - | - | 0.078 | 0.216 | 0.398 |
| | DUSt3R†‡ | 0.301 | 0.495 | 0.657 | 0.085 | 0.210 | 0.398 | 0.166 | 0.304 | 0.437 |
| | MASt3R†‡ | 0.372 | 0.561 | 0.709 | 0.083 | 0.200 | 0.381 | 0.234 | 0.396 | 0.541 |
| | RoMa†‡ | 0.546 | 0.698 | 0.797 | **0.168** | **0.361** | **0.575** | 0.463 | 0.588 | 0.689 |
| *PnP+RANSAC only* | NoPoSplat | 0.572 | 0.728 | 0.833 | 0.078 | 0.198 | 0.394 | 0.337 | 0.497 | 0.646 |
| | Ours (2DGS) | 0.588 | 0.737 | 0.832 | 0.085 | 0.223 | 0.432 | 0.344 | 0.513 | 0.659 |
| | Ours (2DGS+Align) | 0.621 | 0.760 | 0.849 | 0.123 | 0.279 | 0.471 | 0.382 | 0.540 | 0.674 |
| | Ours (2DGS+Orient) | 0.613 | 0.756 | 0.848 | 0.118 | 0.267 | 0.460 | 0.376 | 0.537 | 0.673 |
| | Ours (2DGS+Align+Orient) | 0.627 | 0.766 | 0.855 | 0.135 | 0.289 | 0.479 | 0.392 | 0.547 | 0.679 |
| *w/ Photometric Optimisation* | NoPoSplat | 0.672 | 0.791 | 0.868 | 0.109 | 0.256 | 0.463 | 0.456 | 0.593 | 0.705 |
| | Ours (2DGS) | 0.672 | 0.788 | 0.859 | 0.129 | 0.298 | 0.515 | 0.460 | 0.599 | 0.713 |
| | Ours (2DGS+Align) | 0.686 | 0.799 | 0.870 | 0.136 | 0.311 | 0.512 | 0.474 | 0.607 | 0.718 |
| | Ours (2DGS+Orient) | 0.679 | 0.798 | 0.871 | 0.141 | 0.323 | 0.520 | 0.475 | 0.610 | 0.721 |
| | **Ours (2DGS+Align+Orient)** | **0.689** | **0.804** | **0.876** | 0.156 | 0.334 | 0.539 | **0.488** | **0.619** | **0.726** |

## 4 Experiments

**Datasets and implementation details.** Following [2, 3, 51], we train our models on the large-scale RealEstate10K [57] (RE10K) dataset, with the train-test splits used by [51]. RE10K comprises predominantly indoor real-estate videos from YouTube, containing 67,477 training and 7,289 testing scenes, with camera poses computed using COLMAP [34]. For evaluating generalization, we further test on two additional datasets: ACID [30], containing aerial nature scenes captured by drones (with COLMAP-computed poses), and ScanNet [5], an RGB-D indoor scene dataset with distinct camera motion and characteristics. Specifically, we evaluate relative pose and geometry estimation on the ScanNet. Our training broadly follows recent generalizable splatting methods; full details are in supplementary material. Code and models will be released.

### 4.1 Relative Pose Evaluation

Relative pose is evaluated by computing the AUC of the cumulative pose error curve at three thresholds. We report results deploying a PnP + RANSAC algorithm to align the Gaussian means

Table 2: **Depth estimation for novel views on ScanNet-V1 [5].** We use the novel-view rendered depth accuracy as a holistic measure of 3D reconstruction and interpolation. Our method outperforms all competitors on every metric. Best scores are in **bold**, and top results *without* pose refinement are underlined. Pose-required methods are marked †.

| | Pose-required† | | | | *w/o* Pose Refine. | | | | | *w* Pose Refine. | | | | |
|---|---|---|---|---|---|---|---|---|---|---|---|---|---|---|
| Metric | PixelSplat (3DGS) | MVSplat (3DGS) | DepthSplat (3DGS) | NoPoSplat (3DGS) | Ours (2DGS) | | | | NoPoSplat (3DGS) | Ours (2DGS) | | | |
| | | | | | $\lambda_a, \lambda_o = 0$ | $\lambda_o = 0$ | $\lambda_a = 0$ | $\lambda_a, \lambda_o \neq 0$ | | $\lambda_a, \lambda_o = 0$ | $\lambda_o = 0$ | $\lambda_a = 0$ | $\lambda_a, \lambda_o \neq 0$ |
| Abs Rel ↓ | 0.299 | 0.189 | 0.135 | 0.131 | 0.121 | 0.109 | 0.115 | 0.108 | 0.126 | 0.114 | 0.102 | 0.106 | **0.100** |
| $\delta_1 < 1.10$ ↑ | 0.552 | 0.412 | 0.578 | 0.554 | 0.668 | 0.679 | 0.674 | 0.680 | 0.567 | 0.692 | 0.706 | 0.704 | **0.707** |
| $\delta_1 < 1.25$ ↑ | 0.818 | 0.745 | 0.864 | 0.851 | 0.879 | 0.890 | 0.884 | 0.892 | 0.861 | 0.884 | 0.901 | 0.898 | **0.904** |

Table 3: **Depth estimation for source views on ScanNet-V1 [5].** Best self-supervised scores are in **bold**, and top results *without* pose refinement are underlined. Pose-required methods are marked †, and those using extra supervision (e.g., ground-truth depth) are marked ‡ (upper-bound reference).

| | Supervised‡ | Pose-required† | | | | Pose-free *w/o* Refine. | | | | | Pose-free *w* Refine. | | | | |
|---|---|---|---|---|---|---|---|---|---|---|---|---|---|---|---|
| Metric | DUSt3R | pixelSplat (3DGS) | MVSplat (3DGS) | DepthSplat (3DGS) | NoPoSplat (3DGS) | Ours (2DGS) | | | | NoPoSplat (3DGS) | Ours (2DGS) | | | |
| | | | | | | $\lambda_a, \lambda_o = 0$ | $\lambda_o = 0$ | $\lambda_a = 0$ | $\lambda_a, \lambda_o \neq 0$ | | $\lambda_a, \lambda_o = 0$ | $\lambda_o = 0$ | $\lambda_a = 0$ | $\lambda_a, \lambda_o \neq 0$ |
| Abs Rel ↓ | 0.059 | 0.288 | 0.132 | 0.105 | 0.121 | 0.118 | 0.111 | 0.114 | 0.109 | 0.112 | 0.105 | 0.100 | 0.102 | **0.098** |
| $\delta_1 < 1.10$ ↑ | 0.886 | 0.553 | 0.641 | **0.722** | 0.662 | 0.665 | 0.672 | 0.671 | 0.675 | 0.698 | 0.705 | 0.714 | 0.713 | 0.716 |
| $\delta_1 < 1.25$ ↑ | 0.967 | 0.820 | 0.891 | **0.914** | 0.869 | 0.875 | 0.886 | 0.881 | 0.888 | 0.883 | 0.894 | 0.904 | 0.900 | 0.907 |

| Inputs | pixelSplat | MVSplat | NoPoSplat | DepthSplat | Ours | GT RGB |
|--------|-----------|---------|-----------|------------|------|--------|

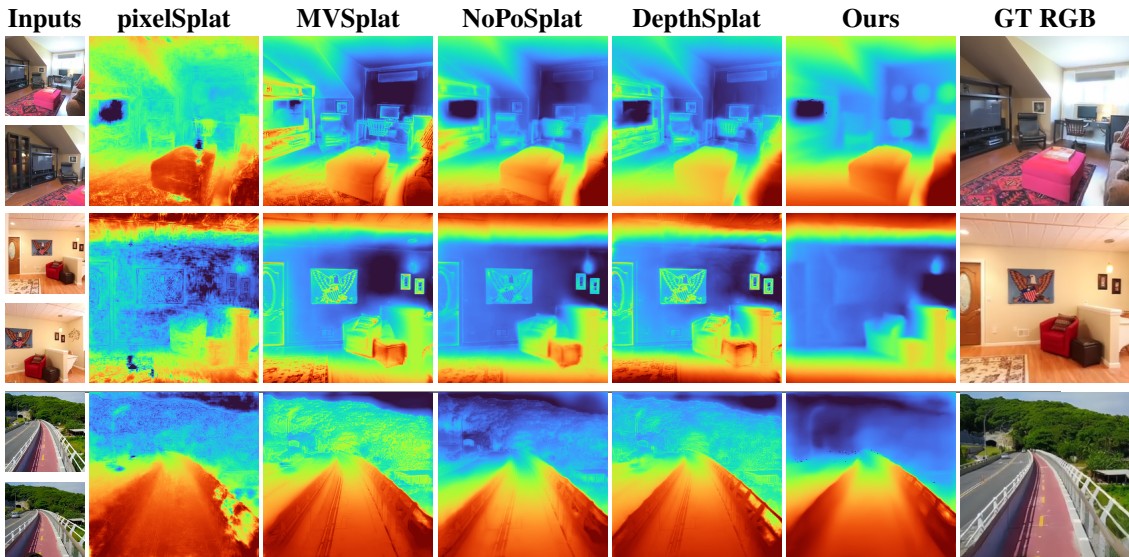

Figure 3: **Qualitative comparison of rendered novel-view depth on RE10k [57] (top two rows) and ACID [30] (bottom row).**

| Inputs | pixelSplat | MVSplat | NoPoSplat | DepthSplat | Ours | GT Depth |
|--------|-----------|---------|-----------|------------|------|----------|

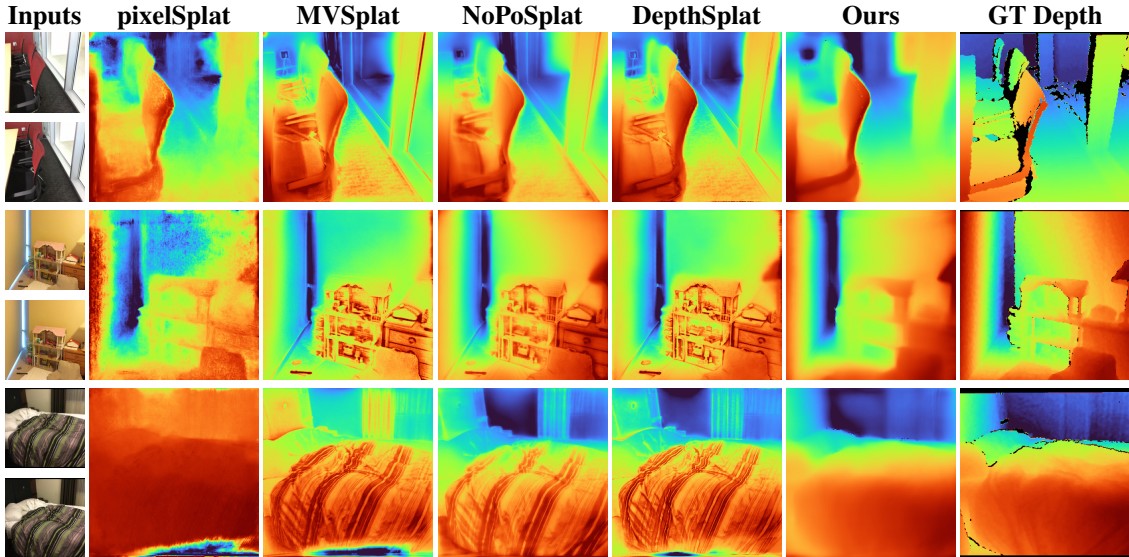

Figure 4: **Qualitative comparison of rendered novel-view depth on ScanNet-V1 [5].**

with image grid as proposed in DUSt3R[42]. NoPosplat [51] proposes a gradient-descent relative pose refinement in which the predicted Gaussians are rendered to generate optimal input image pairs for refining the pose obtained by PnP+RANSAC. Pose Jacobians from [31] are used for this refinement over a fixed number of iterations. Since these Jacobians expect 3D Gaussians, we lift our 2D Gaussians by assigning a small nonzero third scale, to facilitate comparisons.

Table 1 compares relative pose estimation across several methods. CoPoNeRF [17] is trained on RE10K and ACID with explicit pose supervision; DUSt3R [42] uses indoor RGB-D and Internet SfM data (e.g., ScanNet++ [52], MegaDepth [28]) with a 3D regression loss supervising both depth and pose; MASt3R [25] follows DUSt3R's scheme but adds large-scale outdoor sequences (Waymo [36]); RoMA [6] is trained on MegaDepth and ScanNet with depth-and-pose supervision. In contrast, our models use no explicit depth supervision and are trained only on RE10K data. Despite this, we outperform all these methods by a large margin, both on the in-domain RE10K test set and in

zero-shot evaluations on ACID and ScanNet. The sole exception is RoMA [6] on ScanNet-V1, the dataset it was explicitly trained on for relative pose estimation.

Compared to NoPoSplat, our method yields substantial relative-pose gains using only PnP+RANSAC. Incorporating the alignment loss $\mathcal{L}_{align}$ produces marked improvements in both in-domain and cross-domain zero-shot tests, while the orientation loss $\mathcal{L}_{orient}$ provides a further pose-estimation boost, with the combined full loss achieving the best performance. As in NoPoSplat, minimizing the input-image synthesis loss also benefits our pose estimation. Although gains on RE10K and ACID were modest, we observe proportionally larger improvements on ScanNet-V1 with this optimization.

## 4.2 Geometry Evaluation

Geometric veracity of the estimated 2D/3D Gaussian splats is the key focus of this work. Traditionally, the geometry predicted by feed-forward neural networks is evaluated by measuring the depth errors for the input views. However, input depths do not capture the interpolation capability of predicted Gaussians and are insensitive to the opacity, orientation, and scale. We propose a more holistic evaluation of the predicted scene structure by rendering multiple virtual depth maps from the reconstructed Gaussians and reporting Absolute Relative Error and depth accuracy for two different thresholds. As we do not aim to extrapolate beyond the given view frustum, we use the same view-synthesis test set for depth evaluation. Virtual depth maps are rendered using the ground-truth relative pose w.r.t the first input frame, assuming perfectly aligned multi-view Gaussians. This puts pose-free methods at a severe disadvantage – small pose alignment errors amplify depth errors – yet they outperform pose-aware counterparts by a large margin, as shown in Table 3. Our approach substantially outperforms the NoPoSplat baseline. While the 2DGS parameterization improves the accuracy of structure recovery, the orientation and alignment losses deliver much larger gains. This trend persists even after per-image pose optimization is performed to align the recovered 3D Gaussians isolating pose-estimation error from reconstruction quality.

We compare novel-view depth estimation of our method against pixelSplat [2], MVSplat [3], NoPoS-plat [51] and DepthSplat [47] in Figure 3 on RE10K and ACID. MVSplat, DepthSplat and NoPoSplat depths are hypersensitive to texture, while pixelSplat produces notably noisier depths in textureless regions. In contrast, our method yields more plausible depths despite not requiring relative poses. Similar trends are observed in Figure 4 on ScanNet test scenes.

While our primary goal is to predict pixel-aligned, geometrically consistent Gaussians for novel-view depth rendering, we also benchmark source-view depth estimation accuracy of all baseline methods in Table 3. For each method, we report their best depth—whether rendered from Gaussians or predicted by their depth-estimation head—under its best-performing configuration. For example, pixelSplat attains its highest accuracy using rendered depth, whereas MVSplat and DepthSplat perform best with their network-predicted depths. Our method achieves the lowest AbsRel error and performs competitively in thresholded accuracy, slightly trailing DepthSplat [47]. More detailed results for one- and two-view depth prediction are provided in the supplementary material.

### 4.2.1 Novel View Synthesis Evaluation

While not central to our contributions, we evaluate in-domain novel-view synthesis against relevant baselines. Our method outperforms prior work in novel-view synthesis on Re10K dataset, largely thanks to its warping-free formulation. We also observe improvements over NoPoSplat when training with our proposed loss. Detailed results in the supplementary material.

## 5 Conclusion

We propose a novel self-supervised, generalizable splatting network that mitigates geometric incon-sistencies in Gaussian splat recovery previously overlooked by the community. Our model produces state-of-the-art, geometrically consistent Gaussian splats from just two unposed images. While we train on RE10k using an asymmetric transformer architecture under self-supervision, our core contributions are invariant to these design choices. The priors introduced here will help future work on generalizable splatting and learning-based 3D scene recovery.

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
