# OpenReview forum: "Geometrically Consistent Generalizable Splatting"
_NeurIPS.cc/2025/Conference — Submitted to NeurIPS 2025_

### Official Review · Reviewer_NUzX · 2025-06-04

**Clarity:** 3
**Significance:** 2
**Originality:** 2
**Rating:** 3
**Confidence:** 4

**Summary:**

In this work, the authors first analyze the inherent ambiguities in the feed-forward Gaussian Splatting problem. They convert 3DGS to 2DGS and then introduce Align Loss and Orient Loss to enhance the consistency between Gaussian primitives. Experimental results show that their method achieves improved geometric consistency in reconstruction, more accurate relative pose estimation, and better novel view synthesis performance.

**Questions:**

1. Could the authors provide the 3DGS ply files and rendering results instead of only showing mesh reconstructions? Because 2DGS performs better than 3DGS in terms of surface reconstruction quality.
2. Why do the rendered depth maps appear less sharp compared to those from NoPoSplat and DepthSplat?
3. I think pixel-wise Gaussian primitives may be overly redundant, and introducing more views significantly increases GPU memory usage. However, in this work, pixel-wise Gaussians seem crucial, likely because the method relies on PnP for pose estimation.

**Ethical Concerns:**

["NO or VERY MINOR ethics concerns only"]

**Final Justification:**

The rebuttal not fully address my concerns. I choose to keep my score as 3.

**Limitations:**

Please see the weaknesses.

**Quality:**

3

**Strengths And Weaknesses:**

Pros:

1. The analysis of the current feed-forward Gaussian Splatting problem is thorough and impressive.
2. The authors introduce *Align Loss* and *Orient Loss* to enhance the consistency between Gaussian primitives, which appear to be highly effective.
3. The experiments are comprehensive and strongly support the authors’ claims.

Cons:

1. I believe this paper lacks novelty, despite achieving state-of-the-art performance in geometric reconstruction and relative pose estimation. The *Orient Loss* is a common strategy in SDF-based methods, similar to the curvature loss used in Neuralangelo. The *Align Loss* essentially functions as a form of bundle adjustment that optimizes the reprojection loss, leading to improved pose estimation. However, I am concerned that the *Align Loss* may significantly slow down training and may not scale well to multi-view settings. Additionally, are the camera extrinsics and intrinsics used in the *Align Loss* also solved via PnP?
2. I find the pose estimation comparison with DUSt3R, MASt3R, and RoMa to be unfair. Since RE10K is not part of their training datasets, their results represent zero-shot performance, which makes the comparison less meaningful.
3. I would like to know the training time for this method. In the supplementary material, the authors mention using 24 NVIDIA A100 GPUs, whereas NoPoSplat achieves training in 6 hours with only 8 GPUs.
4. The novel view synthesis results shown in Table 4 of the supplementary material seem unusual. Why does 2DGS outperform 3DGS? Additionally, the *Orient Loss* appears to have little impact on novel view synthesis quality. Why does the combination of *Align Loss* and *Orient Loss* result in better rendering quality? I also suggest including qualitative comparisons in the novel view synthesis setting. Furthermore, what is the NVS performance of DepthSplat?
5. I believe the paper should also include comparisons with FLARE, which is a closely related work.
6. The coordinate conventions for normals in MVSplat and NoPoSplat appear to differ from those used in this work. This should be clarified.
7. No demo was submitted.

---

> ### Author Rebuttal · Authors · 2025-07-30
>
> **Novelty and losses**
>
> We believe that the reviewer missed the central premise of our work, which challenges the status quo in the important field of learning to generate splats from images. The view synthesis loss, used prevalently in the field, is shown to be ill-posed in estimating over-parameterised splats and needs to be amended. In the absence of the insights from this paper, the community will continue to throw more data onto the problem or explore the next set of architectural changes that provide minor improvements in terms of avoiding structural degeneracies in self-supervised splatting.
>
>
> The reviewer cites Neuralangelo’s regularisation, which penalises estimated curvature from the SDF alongside an eikonal constraint, to establish that the proposed orientation loss is not novel. We argue that splats are not trivial to convert to SDF. Further, our regularisation does not estimate or penalise curvature at all. We thus wonder if the reviewer misinterpreted critical aspects of the presented approach.
>
> To clarify, \\(L_{orient}\\) is a consistency term between two different normal estimates coming directly from Gaussians in a rendering-free manner. The local normal of the surface is estimated (i) from Gaussian means using image neighbourhoods, and these are forced to align with (ii) the orientation of the Gaussians coming from covariance matrices as defined in 2DGS. This simple construct (that the authors have not seen used in Gaussian-splatting literature at all) provides a necessary signal for orientation supervision for the first time.
>
> \\(L_{orient}\\) is even simpler than the reviewer assumed. It follows the most natural way of estimating local surface normals from a depth map, as done in creating the NYUD-v2 dataset. Our submission is that this simple loss is stable and crucial for learning orientations. The well-celebrated loss introduced in 2DGS is complex in design, involves alpha composition of orientations, is slower to compute, and has been shown to be unstable. We believe that unnecessary complexity should not be encouraged for the sake of being novel.
>
> \\(L_{align}\\) enforces a bundle-adjustment-style loss that the reviewer alludes to—though it’s minimised only at context-view locations for which Gaussian means are predicted and is computationally efficient.
> *(Note: similar to all generalizable splatting approaches, pre-estimated intrinsics and extrinsics are used during training, so no PnP is solved in this loss.)*
> DUST3R did not require such a loss as it had full 3D ground truth for training. Our work shows that when ground-truth depths are not available for learning, this natural drop-in replacement becomes essential in adapting DUST3R-like frameworks self-supervised—a trick NoPoSplat and all other variants of DUST3R have missed to date.
>
> These contributions are by no means trivial or insignificant. Our paper isolates some critical oversights by several works that extended depth and point-cloud estimation frameworks to splat predictions. Ignoring the degeneracies explained leads to meaningless estimations of two-thirds of the splat parameters by all splat predictors, as shown in Figure 2. We believe these findings are crucial for meaningful development in the field of generalizable splatting.
>
>
> **Pose estimation comparison with RoMa and DUST3R**
>
> While Re10K results evaluate in-domain performance of our approach and out-of-domain performance for RoMa and DUST3R, the converse is true on ScanNet, on which our approach was not trained but baselines were. ACID remains out of domain for all methods. Note that methods like DUST3R also use more supervision (ground-truth depth maps) and more data, facilitating generalisation. Despite this, our approach outperforms them in several aspects.
>
> We agree that Table 1 places a lot of cognitive load on the reader. However, this pose-evaluation setup (and presentation) is standard practice in the field. Our Table 1 mimics NoPoSplat and several other works. As foundation models have evolved, several well-known models are trained on very large and sometimes publicly unavailable datasets. This paper aims to provide sensible ablations justifying the core contribution of the work (risking being called a NoPoSplat variant) while facilitating comparison with the state of the art as fairly as possible.
>
> **Training time**
>
> We want to highlight that our inference is marginally faster than NoPoSplat (estimating one fewer scale); losses are also computed at inference.
>
> During training, Align and Orient losses add negligible overhead compared to rendering and backprop; both losses are local and fairly easy to estimate requiring projections of Gaussians means to image place. Fixing training hyperparameters, hardware and the rasterizer (2DGS), total training times for the models are:
>
> | Loss            | Time (min)|
> |-----------------------------|-----------|
> | view-synthesis only  |  335.8  |
> | + \\(L_{orient}\\)        |  335.7  |
> | + \\(L_{orient}\\) + \\(L_{align}\\) |  339.3  |
>
> Variations are statistically insignificant. We will add these in supplementary material. Note that we use A100 GPUs (40GB), which only has half the GPU memory compared to hardware used in NoPoSplat. This has implications on the total number of GPUs used to have large batch size (similar to NoPoSplat) and requires some changes in training hyper-parameters, which are reported. We reproduced NoPoSplat results with our hyperparameters and hardware to ensure fair comparison and reported the same in supplementary material. The discrepancy in training time or number of GPU is only due to the hardware restrictions.
>
> **Novel view synthesis results**
>
> The focus of this work is not to improve novel view synthesis, and no explicit design choices provide any meaningful change in view-synthesis quality. Changes in view synthesis due to different loss functions are statistically insignificant and should not be given much emphasis at all in our opinion. However, view-synthesis experiments were included in the supplementary document for completeness, ensuring geometric consistency does not come at the cost of view-synthesis capability as is the case for some splatting methods like 2DGS.
>
>
>
> **Coordinate conventions**
>
> Please note that MVSplat/pixelSplat does not prescribe any surface-normal estimation procedure or conventions. The so-called “Normals” for all baselines are visualised by us following the surface-normal definition in the 2DGS/PGRS papers: the minimal-elongation eigenvector (corresponding to the lowest eigenvalue) at every context-image pixel. These visualisations are used to highlight that baseline orientations lack any coherent structure and that estimated scales do not conform to underlying geometry. Alternatively, 3D Gaussians can also be visualised as larger opaque blobs, but we believe our image-like visualisations are more effective and easier to interpret (Figure 2). These visualization choices, however, do not affect the reported baseline results in this paper.
>
> **Alternative visualization and quality of results**
>
> As suggested, we can share PLY-file visualizations on the project page; however, these are highly sensitive to viewpoint choice. Rendered Gaussians provide a qualitative evaluation of view-synthesis performance, and many geometric errors and inconsistencies get disguised in such visualizations. We choose a mesh visualization (and standard depth visualization) which highlights geometric accuracy/consistency, or lack thereof.
>
> **Lack of sharpness**
>
> We refer the reviewer to our earlier responses. While a geometric edge-aware regularisation can be trivially used to improve the situation to some extent, our current results are qualitatively and quantitatively a significant improvement over the baselines without these modifications.
>
> **FLARE comparison**
>
> FLARE is trained on multiple large RGB–D datasets (including ScanNet), uses ground-truth depths for Gaussian means, and adds rendered-depth supervision via AnyDepth. This strong geometric supervision and extra data give FLARE several advantages over our method, which relies only on Re10K and no explicit depth supervision, making direct comparisons inherently unfair and grossly disadvantageous for us and other baselines.
>
> Additionally, FLARE is very sensitive to test-time image resolution, uses five frames during testing, and reports different pose and depth evaluation metrics, which further complicates direct comparison.
>
> That said, we evaluated pose estimation against FLARE using the same number of input views (2), the same metrics (AUC@[5$^\\circ$,10$^\\circ$,20$^\\circ$]), and identical PnP-RANSAC hyperparameters to ensure a fair comparison.
>
> On the ScanNet test set, our method (using a lower resolution) still outperforms FLARE. Note that we have not trained our model on ScanNet, while FLARE has.
>
> | Method | Input resolution | 5$^\\circ$ | 10$^\\circ$ | 20$^\\circ$ |
> |--------|------------------|------------------|-------------------|-------------------|
> | FLARE  | 512×384          | 0.116            | 0.262             | 0.467             |
> | **Ours**   | 256×256          | 0.135            | 0.289             | 0.479             |
>
>
> **Pixel-wise Gaussian**
>
> The association of Gaussians with context-view pixels is a construct that all generalizable splatting algorithms we study adopt. This constraint comes from the fact that many splat predictors append DPT-like decoders to estimate Gaussian parameters on existing depth- or image-aligned 3D point-cloud predictors. The scalability of per-pixel Gaussian prediction is an important open problem and perhaps requires architectural changes. This work restricts itself to making prevalent pixel-aligned generalizable splatting geometrically accurate. We agree that \\(L_{align}\\) assumes pixel alignment and might need to be adapted to suit novel non-pixel-aligned Gaussian representations of the future. Findings from this work will provide a good foundation for such adaptations.

---

> > ### Comment · Reviewer_NUzX · 2025-08-05
> >
> > The rebuttal did not fully address my concerns:
> > 1. In Neuralangelo and this paper, the main idea is similar, that is, the Orient Loss and the curvature loss both aim to make the  geometry more smooth, just the different implementation. I think I fully understand this part.
> > 2. I strongly recommend that the author align the coordinate system of MVSplat/pixelSplat with the coordinate system used in this paper, as this will facilitate understanding of the paper.

---

> > > ### Author Response · Authors · 2025-08-08
> > >
> > > Reviewer had many concerns / asked for clarification on many aspect of our work.
> > > We hope that we addressed most of the technical queries in our response, in particular regarding training time, fairness of experimental comparisons and lack of sharpness in our results.
> > >
> > > Reviewers comment in light of our rebuttal suggest that the remaining concern is perceived lack of novelty in our work alongside coordinate conventions used.
> > >
> > > As mentioned in the response, we are unaware of any differences in the coordinate systems between MVsplat, pixel-Splat and NopoSplat which we choose to integrate our losses to. We would be grateful if the reviewer can elaborate how our coordinate conventions are misaligned form the mentioned approaches so we can aim to address this issue and improve our work.
> > >
> > > We agree that our approach uses geometric losses inspired from the vast literature of scene modelling but for the first time in context of predicting splats. Such losses were shown to be essential to make an important learning problem well posed. We provided first feed-forward 3D splat predictor that predicts non-degenerate orientations and scales and  have state of art pose and structure estimation performance without any compromise on view synthesis.
> > >
> > > We leave the reviewers and ACs to judge the requirements on mathematical novelty expected at NeurIPS over appreciating well designed, simple learning frameworks that challenge and rectify important design oversights in solving an important problems.
> > >
> > > We thank reviewer for their insightful comment and suggestions during the review and discussion process.

---

### Official Review · Reviewer_MqDQ · 2025-06-20

**Clarity:** 4
**Significance:** 2
**Originality:** 3
**Rating:** 3
**Confidence:** 3

**Summary:**

This paper introduces a structurally constrained 2D Gaussian Splatting approach to improve orientation stability and geometric consistency in pose-free settings. It replaces the rank-3 covariance in NoPoSplat with a rank-2 version and adds two geometric regularizers: a pixel alignment loss and a directional consistency loss. The method outperforms existing approaches on RE10K, especially in normal consistency and pose estimation tasks.

**Questions:**

1. The authors should clarify why geometric losses like L_orient and L_align cannot be applied directly to full-rank 3DGS. If these losses are equally effective and stable in the original representation, then rank reduction is unnecessary and the design motivation becomes unconvincing.
2. Does rank reduction lead to a loss of expressiveness—for example, the inability to model volume, thickness, or multi-normal regions? Has the impact of this on downstream tasks (e.g., relighting or reconstruction in fuzzy areas) been evaluated? The paper does not address this and should include relevant discussion.
3. The paper claims improved training stability with rank reduction but provides no supporting efficiency data, such as memory usage, parameter count, or convergence speed. Without such evidence, the rank-2 design appears more like a trade-off than an actual advantage.
4. Has the idea of using a mixed-rank strategy been considered? For instance, applying rank-2 in structurally clear regions and retaining rank-3 where uncertainty is high.

**Ethical Concerns:**

["NO or VERY MINOR ethics concerns only"]

**Final Justification:**

The authors did not address my concerns regarding the motivation behind the low-rank (2DGS) design. I believe the current manuscript does not provide a sufficiently clear or theoretically grounded explanation for adopting the low-rank strategy, relying primarily on empirical results rather than in-depth theoretical derivation or mechanistic analysis.

**Limitations:**

It is recommended to explicitly include a “Limitations” section.

**Paper Formatting Concerns:**

No major issues found.

**Quality:**

3

**Strengths And Weaknesses:**

The paper focuses on structure modeling in pose-free and generalizable settings, with a clear problem setup. By reducing the Gaussian covariance to rank-2 and introducing two geometric losses (L_align and L_orient), the method achieves solid performance on tasks like structure reconstruction, PnP pose estimation, and normal fusion. Training is stable, and the visual analysis in the appendix adds some credibility.

However, the method itself is a relatively minor modification of NoPoSplat, with limited novelty. The rank reduction improves stability but also sacrifices expressiveness, making the model unsuitable for volumetric structures or multi-normal regions. More importantly, the paper doesn’t justify why these geometric losses can’t be applied directly to full-rank 3DGS. If they can, the rank-2 design becomes unnecessary. In addition, there’s no discussion of memory usage, efficiency, or applicability limits, leaving important questions unanswered.

---

> ### Author Rebuttal · Authors · 2025-07-30
>
> We reiterate that the paper highlights important degeneracies posed in predicting 3D Gaussians using feed-forward networks. Despite several efforts, these degeneracies persist in the literature and are explained to be due to the insufficiency of view-synthesis losses. We present a thorough analysis of how existing frameworks converge to predict meaningless Gaussian scales and orientations, motivate and implement geometric regularisations to estimate meaningful surfel-like 2D Gaussians via feed-forward neural networks. The contributions of the presented approach are adaptable to any existing pose-free or pose-aware feed-forward splat predictors trained with or without depth supervision. In our view, this work would generate immediate impact and increase the applicability of splat predictors beyond the view-synthesis application. We encourage the reviewer to read the comments to Reviewer S4Ch and Reviewer ugeC, where the novelty and potential impact of our framework are explained in detail. We attempt to address other specific questions of the reviewer here:
>
>
> **\\(L_{orient}\\) , \\(L_{align}\\) and full rank 3D Gaussian**
>
> We want to clarify that \\(L_{orient}\\) or \\(L_{align}\\) loss can be applied to 3D Gaussians with full-rank covariance as well. \\(L_{align}\\) operates solely on the 3D Gaussian means, so it can be used as is if the predictions are full 3D covariance matrices. \\(L_{orient}\\) merely enforces two different estimates of local surface normals to be consistent. One local-normal estimate relies on image-based neighbourhood and Gaussian means, imposing no restriction on the rank of the covariance matrix. The alternative local normal is defined by the eigenvector of the covariance matrix corresponding to the smallest eigenvalue—they are estimable for both degenerate and full-rank Gaussian covariance.
>
> In fact, the non-degenerate covariance matrices and the normals estimated from them as prescribed above are used in the work PGSR: Planar-based Gaussian Splatting for Efficient and High-Fidelity Surface Reconstruction. This work advocates replacing the hard surfel-like constraints used in 2DGS with a soft penalty term that encourages low-rank Gaussian covariance where possible. This is very similar to the “mixed rank” strategy the reviewer mentioned and, in fact, was considered during the conception of the paper but rejected based on empirical evidence.
>
> In the original 2DGS work, rank-deficient covariance matrices are shown to improve geometric accuracy of the estimated Gaussians, although this comes at the cost of view-synthesis performance degradation, i.e., 2DGS provides more accurate meshes but trails behind 3DGS in view-synthesis accuracy. PGSR proposes to take a middle ground by penalising the rank of the Gaussian covariance to improve view synthesis. It thus can be argued that the expressiveness of full-rank Gaussians is beneficial, at least for view synthesis.
>
> However, contrary to findings of 2DGS and PGRS, it was found in our experiments that, in the context of learning splat predictors, 2DGS outperforms 3DGS representation on all grounds, including accurate view synthesis. This is perhaps due to the fact that overly expressive 3DGS can easily overfit to the texture of individual scenes—leading to better novel-view synthesis for one scene. However, these improvements are scene-specific and unstructured when generalised to large datasets. In the context of learning, 3DGS’s added expressiveness makes the loss function susceptible to local minima. Full ablations of pose (Table 1), depth (Table 2), and view-synthesis experiments (supplementary material) suggest that the expressiveness of full covariance is always harmful in our setup.
>
> We agree that imposing hard rank deficiency reduces theoretical expressiveness. However, more expressiveness does not guarantee better results. On the contrary, over-expressive representations are sometimes the source of degeneracies, invoking the need for strong regularisations. Full-rank/mixed-rank Gaussians were rejected based on our experimental results, and we will explain this in the paper more clearly. However, we are happy to include an ablation for 3DGS with our loss for completeness and will add the relevant discussion.
>
>
> **Other application / evaluation of expressibility**
>
> We do not understand what the reviewer means by the 2DGS’s “inability to model volume, thickness, or multi-normal regions”. We are unaware of the inability of 2DGS to model thickness or volume. Perhaps the reviewer means transparent regions when they mention fuzzy areas and normal discontinuities when they mention multi-normal regions? Note that the rendered depths were evaluated on all pixels of context as well as virtual views on the ScanNet dataset (might these pixels include such fuzzy and multi-normal areas). We are happy to evaluate the trained models with the reviewer’s prescribed metrics on selected pixels to address the concern. We will be grateful if the reviewer will be kind enough to elaborate and suggest a concrete evaluation protocol that answers these concerns.
>
> Relighting and other applications of 3D Gaussian splats are out of the scope of this paper and are not discussed in any feed-forward splats either. We inherit all the limitations and advantages from the 2DGS and generalizable splatting in terms of deploying the predicted splats to such applications. The limitations inherited from literature will be listed in the paper.
>
>
> **Training stability due to 2DGS**
>
> We are unaware of any claim the paper makes regarding training stability due to 2DGS. 3DGS variants (NoPoSplat) are trained successfully without any stability issues in this work. We, however, claim geometric accuracy when deploying 2DGS. We also mentioned that the 2DGS paper’s original normal loss was found unstable during training and \\(L_{orient}\\) was introduced. Any text suggesting that 2DGS brings training stability should be removed from the paper, and we appreciate the reviewer for identifying such a mistake.
>
>
> **View synthesis ablations**
>
> The paper’s focus is on accurate geometric estimation, and it does not discuss a minor reduction in the number of parameters (it trivially estimates two instead of three scales, keeping all other architecture and prediction heads of NoPoSplat intact). Predicting one less scalar provides a minor efficiency boost that, in our view, is insignificant to discuss. We fail to see a trade-off in using 2DGS over 3DGS. 2DGS-parameterisation is theoretically sound, leads to minor efficiency gains, and provides boost in accuracy in all aspects of evaluation that we performed over 3DGS counterpart—including view synthesis.

---

> > ### Comment · Reviewer_MqDQ · 2025-08-05
> >
> > Thank you for your reply, but it did not address my concerns regarding the motivation behind the low-rank (2DGS) design. I believe the current manuscript does not provide a sufficiently clear or theoretically grounded explanation for adopting the low-rank strategy, relying primarily on empirical results rather than in-depth theoretical derivation or mechanistic analysis. I strongly suggest the authors further clarify the necessity and theoretical rationale for the low-rank design, explicitly discussing the trade-offs in terms of expressive power, optimization convergence, and practical application. If the motivation remains primarily empirical, it may weaken the methodological novelty and scientific credibility of the work. Please consider strengthening the analysis of the core motivation and providing quantitative or theoretical support. Additionally, I recommend including ablation studies on the relevant design choices and presenting concrete case analyses and visualizations on the ScanNet dataset to enhance the rigor and practical value of the paper. It is also important to clarify which scenes from ScanNet are selected in this study.

---

> > > ### Author Response · Authors · 2025-08-09
> > >
> > > We appreciate constructive feedback that suggests experimental rigour and request the mechanistic understanding of the proposed pipeline. While most reviewers lauded the presented experimental evaluation and motivation of the paper, we believe that some additional analysis might be helpful - especially for readers who are non-experts in the 3D reconstruction domain. In the absence of any particular suggestions from the reviewer's side, we propose to enhance the supplementary material with two minor (but hopefully effective) ways:
> > >
> > > **Adding visualisations:** While a full quantitative ablation of our regularisers is presented in the paper, perhaps visualising misalignment in estimated Gaussian means (in the absence of the alignment loss) and degenerated orientations in the absence of orientation loss will help readers understand the need and impact of these losses visually. Note that these losses were motivated for addressing these two issues that the current state of the art suffers from. We thought that Figure 2 (alongside quantitative ablations) was sufficient in establishing the need and impact of proposed losses. However, visualising the impact of each loss term separately might improve clarity/ rigour.
> > >
> > > **Adding one extra ablation:**  To address the concern regarding lack of expressiveness in rank-deficient Gaussians, we will include one extra experiment where full-rank Gaussians are learned with the proposed orientation and align loss. Note that such a full rank formulation was already shown to be unhelpful in the absence of proposed losses. This indicated that the expressiveness of full rank Gaussians was not useful in improving performance on any metric we studied. However, additional experiments such as this will hopefully provide (a) an assurance that the losses presented can be trivially used with full rank formulations and (ii) conclusively establish that the expressiveness of full rank Gaussians provides no advantage even with the introduced prior.
> > >
> > > We reiterate that our work does NOT introduce rank-deficient Gaussians nor depend on them. Proposed losses are trivially adaptable to predict full rank 3D Gaussians for whatever reason one wants to choose them. We thought this should have addressed the primary concerns of the reviewer. We are surprised that the reviewer advocates including theoretical derivations in this regard to warrant publication. The reviewer does not mention what these theoretical derivations should be, nor how they must differ from anything already presented in the 2DGS paper.
> > >
> > > We had the option to work with full or rank-deficient Gaussians. We experimented with both, without any prejudice and presented experimental findings. Our experiments suggest that while the original rank-deficient 2DGS formulation was reported to have sacrificed per-scene view synthesis performance to gain better geometry, no such trade-offs were found in our case. We respectfully disagree that relying on experimental evidence in choosing one representation over another weakens the novelty or scientific credibility of our work. Ignoring experimental evidence to advocate any theory is, on the other hand, unscientific; regardless of its mathematical elegance, novelty or sound derivations provided in support of the theory.
> > >
> > > We leave judging novelty, complexity and need for theoretical derivations to get a work published in a prestigious conference as NeurIPS, to reviewers and the ACs' wisdom.
> > >
> > > We sincerely thank the reviewer for their time and consideration.

---

### Official Review · Reviewer_ugeC · 2025-06-25

**Clarity:** 3
**Significance:** 2
**Originality:** 2
**Rating:** 2
**Confidence:** 4

**Summary:**

In this paper, the authors stated that a view-synthesis loss is not enough. To address this, after analysis, the authors add two additional regularization losses: orientation loss and alignment loss.

**Questions:**

1. In Line 9, the authors claimed their goal is to learn pose-free generalizable splatting, which means no pose provided. But in Line 221, the authors said, 'with known camera extrinsics'. This makes me confused.
2. Following question 1, if there is no GT pose involved during training, that is fine. If the 'known poses' are actually used during training, I would like to see the comparisons with other feed-forward reconstruction models using Gaussian Splatting supervised by GT poses during training like Splatt3r.

I will consider increasing my ratings if the authors can answer my questions reasonably.

**Ethical Concerns:**

["NO or VERY MINOR ethics concerns only"]

**Final Justification:**

I kept my rating, since my concerns were not addressed. These concerns are mostly shared with other reviewers.

**Limitations:**

Please refer to weaknesses.

**Paper Formatting Concerns:**

None.

**Quality:**

3

**Strengths And Weaknesses:**

**Strengths**
1. The authors analyze the veracity and geometric meaning of the Gaussian orientations and elongations.
2. The authors conducted extensive experiments to support their statements.

**Weaknesses**

1.This paper is somehow lacking novelty and contributions. The core parts are those two additional regularization terms, however, regarding the first loss, it shares similar high-level ideas with the ones in 2DGS, but with some technical adjustments. My questions on the second loss are in the following cells.

2. Typos in Line 210.

3. The result improvement is somehow tiny. In Figure 3 and 4, I understand that the depth should not be affected by the texture, but it seems the results from the proposed method is blurrier around the border regions.

I will consider increasing my ratings if the authors can answer my questions reasonably.

---

> ### Author Rebuttal · Authors · 2025-07-30
>
> We thank the reviewer for their valuable time and effort in reviewing our paper. The typos will be fixed, and write-up suggestions will be incorporated.
>
> We disagree with the reviewer on the lack of novelty and contribution in our work. It should be noted that feed-forward multi-view splat prediction is an important problem. This paper highlights that all existing approaches that claim to solve this problem effectively suffer from a lack of supervision, due to which they fail to learn the orientation and scale of Gaussians. Ours is the first framework to highlight these important oversights and fix them in a systematic, generalisable manner.
>
> Of course, relations between point clouds, mesh, and volumetric density fields have been well studied in the literature for decades. We readily admit that our regularisations are inspired—or even “lifted”—from this rich literature. They are simple but effective in achieving our goals. Note that 2DGS itself inherits its surfel-like splat formulation as well as geometric regularisation from this very literature, using them in the context of multi-view splat inference (without learning). We do the same for generalisable splatting (with learning).
>
> **Technical adjustment of 2DGS (\\(L_{orient}\\))**
>
> We request the reviewer to consider that the design and deployment of meaningful regularisations in the context of a specific problem is non-trivial and requires care. For example, our work shows that using regularisation from 2DGS _as is_ does not work (see supplementary material). The predicted splats from such regularisation fail to capture detailed scene geometry, and training diverges to trivial planar reconstructions. This is likely due to the extremely sparse views used for training in our case; the original 2DGS algorithm was never tried under such conditions. The proposed loss, despite its simplicity, is more stable and effective than the well-celebrated 2DGS normal-consistency loss and should not be ignored as a “technical adjustment.” The 2DGS normal-consistency term is relatively complex, requires gradients to be propagated via the rendering engine, and is novel in its formulation—but leads to optimisation instability. Our solution is shown to be empirically better (and essential) and should not be penalised for its simplicity.
>
> Similarly, our alignment loss supplements the inherent structure–pose ambiguity that DUST3R-like frameworks suffer from when depth supervision is unavailable; multiple point-cloud/camera-pose combinations can produce the same rendering. It is obvious in hindsight that alignment loss must be used alongside view-synthesis loss to make the learning objective well posed. However, all generalisable splatting algorithms that attempt to “splatify” DUST3R crucially missed this requirement. Not using it is a critical design flaw of some prevalent frameworks in the literature. We agree that more sophisticated regularisation terms can be invented, and some might outperform our proposed alternative. However, acknowledging that geometric degeneracies exist in the absence of regularisers is the first step toward making meaningful progress.
>
>
> **Tiny improvements and blurry depths**
>
> We want to highlight that duplicating texture to the geometry is neither a minor nor a single issue of existing frameworks. Geometry from existing generalizable splatting methods is unusable for most practical applications, such as navigation, manipulation, or AR/VR. Consider, for example, trying to augment a toy duck on top of the quilt in Figure 4 (last row). Local planes are not readily estimable from such depths for successful augmentation. Even after correctly placing the duck on a local plane on a quilt, the entire quilt texture (stripes) will be assumed to occlude the small toy duck based on the estimated geometry. Alternatively, try to imagine a toy copter attempting to fly through the paintings and walls due to the big holes predicted by baseline approaches. Our “blurrier” geometric edges do not have quite the same adverse effect when used for these tasks.
>
> We do admit that the results are blurred around true geometric edges more than desired. One reason is that \\(L_{orient}\\) was minimised at all the input image pixels. However, the underlying normal-estimation procedure from Gaussian means is undefined at scene edges and should be ignored at discontinuities. An edge-aware variant of discarding \\(L_{orient}\\) at depth discontinuities has since been adapted, and it provides relatively sharper reconstructions. It should be noted that such a change is trivial and does not make the approach any more novel. We maintain that the reconstructions provided in the original manuscript are significantly better than baselines on many grounds, and these improvements are overlooked.
>
> **Pose-free vs pose-aware methods**
>
> The reviewer misunderstood our categorisation of the pose-free generalisable splatting approach. We clarify that *all* methods discussed require ground truth relative poses between input views (and virtual views) during training. The comparisons are fair, as all approaches use ground truth camera poses during training. The term “pose-free” here means that NoPoSplat, Splatt3R, and our approach do not need relative poses at inference; MVSplat and pixelSplat require this information at inference and are thus called pose-aware.
>
> Regarding Splatt3R, the framework uniquely does not modify the registered point-cloud estimation of DUST3R at all. Thus, the point-cloud estimation and PnP-RANSAC–based pose-estimation accuracy of DUST3R remain intact in Splatt3R.
>
> Literature shows that Splatt3R’s view-synthesis performance is inferior to almost all other baselines. We also evaluated its rendered depth, which exhibits the same geometric inconsistencies. Splatt3R’s rendered depths have an average AbsRel error of 0.148 (ours: 0.100) on the ScanNet test set. Note that Splatt3R was trained on ScanNet. We can include these results in the paper upon acceptance.

---

> > ### Comment · Reviewer_ugeC · 2025-08-06
> >
> > Thanks to the authors for the rebuttal. However, it does not address my concerns. The biggest concern I have, which is also shared with other reviewers, is the lack of contribution in this paper. The claimed contributions of this work are highly similar to the existing works, but are implemented in different places. Besides, I highly suggest that the authors conduct a thorough theoretical analysis of the motivations to add these two regularization terms. Regarding performance, with such regularization terms, the improvement is tiny from my perspective. I am inclined to keep my current ratings and discuss with other reviewers and AC later. Thank you.

---

### Official Review · Reviewer_S4Ch · 2025-06-28

**Clarity:** 3
**Significance:** 2
**Originality:** 2
**Rating:** 3
**Confidence:** 4

**Summary:**

This paper tackles the task of “generalizable” gaussian splatting i.e. learning a feedforward predictor that, given a set of unposed images, infers a corresponding gaussian splatting representation. It largely builds on a prior framework (NoPoSplat) for this, but introduces additional regularizations to improve the quality of the inferred geometry: a) modifying 3D gaussians to 2D gaussians, b) encouraging the gaussians predicted by adjacent pixels to have similar orientations, c) enforcing the pixel-wise gaussians to lie along the corresponding ray.

The system is trained on RealEstate10k and evaluated on a couple real world datasets, and it demonstrates some improvements over the base model (NoPoSplat) in terms of the depth, pose, and view synthesis accuracy.

**Questions:**

Could the authors provide some intuition on why there is a large improvement in Table 1 over NoPoSplat in the "PnP + RANSAC” but this larger gains reduces after the photometric optimization?

**Ethical Concerns:**

["NO or VERY MINOR ethics concerns only"]

**Final Justification:**

I'd like to thank the authors for the detailed response, and in particular for correcting my understanding of the orientation loss and highlighting the stronger empirical gains for pose estimation. In light of these, I would be willing to increase my rating slightly but would still not argue for acceptance given the concerns about technical contributions.

**Limitations:**

Yes

**Quality:**

2

**Strengths And Weaknesses:**

Strengths:

+ This paper is well-motivated. The typical pipeline in training ‘generalizable’ gaussian splatting only optimizes for view synthesis loss, leading to under-constrained geometry. The goal in this paper is to overcome this limitation.

+ The ideas presented in the paper are all sensible and well-implemented. The three regularizations added (2D gaussians, local normal consistency, and ray alignment) are all intuitive.

+ The impact of the changes made is ablated well and the experiments show the benefits of adding one regularization component at a time.

Weaknesses:

- The contributions of this work are fundamentally limited to including regularizations in an existing framework. Even more so, regularizations such as these are also common (e.g. local normal consistency is a common term for mesh prediction, encouraging Gaussian to be like surfels is another common idea in recent work) — the contribution here is to merely apply these to the NoPoSplat framework.

- While the paper shows consistent improvements over NoPoSplat, these are relatively marginal e.g. improved 2% accuracy in camera estimation or depth prediction (and similarly minor gains in pose). While this is not an issue on its own e.g. a paper showing a fundamentally new approach that is comparable to or marginally improves SoTA is ok, combined with the limited technical contribution, the marginal empirical gains make the impact of this work rather limited. Basically, the approach here is not very novel and the results are not very significant — and this combination limits the impact of the work.

- The results are only shown on indoor scene-level datasets which predominantly have flat surfaces, but some of the regularizations e.g. local normal consistency maybe harmful in other domains e.g. object-centric datasets with high curvature in geometry. As such experiments on datasets like Co3D or Objaverse would have been very helpful in addition to reassure the reader about the contributions being broadly applicable.

---

> ### Author Rebuttal · Authors · 2025-07-30
>
> We thank the reviewer for their time and effort in reviewing this submission. The reviewer agrees with the motivation and claims made. They also agree that these claims are substantiated through experiments and appropriate ablations. They, however, choose to recommend rejection of the paper, citing the simplicity of the ideas that this paper introduced and underestimating the impact of our work. We provide a detailed thematic response to address the reviewer’s concerns here.
>
>
> **Novelty and “Merely Regularising NoPoSplat”**
>
> We respectfully disagree with the reviewer’s assessment that incorporating multiple well-thought-out regularizations in solving an important learning problem is limiting and not novel enough to be accepted at NeurIPS. On the contrary, this paper challenges the status quo, isolating geometric degeneracies that all existing solutions in the literature suffer from and overlook. The paper proposes to fix these degeneracies systematically with remedies that are trivially adoptable in nearly all state-of-the-art splat predictors—creating a large impact on the field. If these are not traits of a good NeurIPS paper, what will be?
>
> Since the introduction of generalised splatting in pixelSplat (CVPR 24), several attempts have been made in the past two years to solve the problem. These include MVSplat (ECCV 24 oral), Splatt3R (CoRR), NoPoSplat (ICLR 25 oral), and DepthSplat (CVPR 25). Much like the reviewer did for our work, many of these works can be trivialised. MVSplat simply replaces the pixelSplat's epipolar-attention transformers with the well-studied cost-volume-based architecture of UniMatch; NoPoSplat plugs in the DUST3R architecture in MVSplat; and Splatt3R adds a DPT decoder head to a frozen pre-trained DUST3R for estimating additional 3D Gaussian parameters. We disagree with such trivialisation of these important works, including our own. Note, however, that all these highly-received papers claim to learn the orientation, scale, and opacity of 3D splats, they use existing network architectures and standard well-studied view-synthesis losses. All of these frameworks fail to do what they claimed effectively, as shown in this paper.
>
> This paper demonstrates degeneracies in predicting splat parameters using view-synthesis loss—this loss is used by the entire community without exception. So the issues mentioned in the paper are **not** NoPoSplat-specific. These traits are shared amongst all splat-prediction frameworks, including the recent adaptation of VGG-NT for splatting in AnySplat. We want to reiterate that NoPoSplat was merely chosen as the baseline, given its state-of-the-art performance at the time and its ability to work without requiring a relative pose between images at inference time. The losses used in the paper provide a “simple recipe” to facilitate necessary supervision for Gaussian orientations and scales for the first time. Findings of this paper have the potential to change the landscape of feed-forward splatting and should not be viewed as a minor adjustment to NoPoSplat alone.
>
> Authors are unaware of any attempt made to isolate and discuss degeneracies in 3D splat predictions in the last two years, let alone address them by deploying suitable regularizations that the reviewer considers “common”. This alone showcases that the ideas presented in this paper are non-trivial. The simplicity of our frameworks is its core strength and should not be used to trivialise our work’s contributions or potential impact.
>
>
> **Marginal improvement over SOTA (2%)**
>
> The 2% improvement comment is a gross generalisation of our experimental findings and is unfounded. We strongly contest that the quantitative improvements of our approach are marginal.
>
> - **Pose estimation improvements:**
> The regularizations improve pose estimation with PnP + RANSAC by **9.6%** (from 57.2% to 62.7% AUC@\$5^\circ\$ error threshold over the baseline on **Re10K**. The improvement is nearly **73% (7.8% vs 13.5%) on ScanNet**. These results showcase the effectiveness of the proposed regularizations and justify our claims of making predicting geometrically meaningful and accurate **raw** splats. Even *with pose optimisation* with view synthesis loss minimisation, our method predicts accurate poses (with AUC@\$5^\circ\$ error threshold) for ***15.6%** cases against 10.9% of NoPoSplat on ScanNet* — well above 2% the reviewer claims. Note that NoPoSplat, on the other hand, can not retain DUST3R's performance with PnP-RANSAC on ScanNet, let alone improve on it. It is evident from experiments that “splatifying” DUST3R naively does not work.
>
> - **Structure estimation improvements:**
> We would like to point out that the paper provides a boost in relative depth accuracy ($\delta_1 < 1.10\uparrow$) of **22.7%** without (**20.6%** with) view synthesis-based relative pose optimisation over the baseline of NoPoSplat on ScanNet. These improvements are in line with NoPoSplat (30% without and 33% with pose refinement over MVSplat) and are not *minor*.
>
> Most importantly, we would like to highlight meaningless Gaussian orientations, unstructured scales and grossly erroneous view-inconsistent depths with texture-sensitive protrusions that “every” studied approach predicts. These degeneracies are unacceptable, and ours is the only approach that fixes them. We encourage the reviewer to read our response to *Reviewer ugeC*, highlighting the qualitative depth prediction improvement that is attained by our approach as well. We are sure that upon a closer inspection of the qualitative results, both the reviewer and ACs will agree that the current state-of-the-art feed-forward networks do not provide splats with accurate enough geometry to be used in applications like robotic navigation and manipulation. The proposed approach makes large strides in attaining this.
>
>
>
> **Pose estimation and photometric optimisation**
>
> The reviewer asks why view-synthesis loss minimisation does not yield a similar boost as observed in NoPoSplat. Note that view-synthesis loss minimisation is tailored to NoPoSplat: large alignment errors in RAW splat predictions are fixed with this optimisation. The optimisation is cumbersome, involving multiple iterations of time-consuming rendering steps. We use it *as is* for completeness and fair comparison with NoPoSplat’s full pose-estimation pipeline. Possible reasons for limited gain:
>
> 1. View-synthesis loss is not the sole loss minimised during training, making it non‑optimal for pose optimisation. A combination of PnP and view-synthesis loss might be more suitable. Moreover, our approach uniquely produces accurate surface normals as Gaussian orientations; these orientations can inform relative rotation estimation (e.g. by minimising point-to-plane geometric errors for aligning rigid surfaces) but are not used in the current pose-estimation pipeline.
>
> 2. The pose-optimisation routine assumes 3D covariance matrices for Gaussians. Our hard, surfel-like Gaussians must be converted to full 3D Gaussians by adding a small third scale, which may introduce inefficiencies.
>
> Selecting or designing an optimal pose-estimation routine is not the focus of this paper. Our work highlights and removes geometric inconsistencies in splat predictions. Pose-estimation heuristics vary widely in generalizable splatting (and other neural-geometric estimation) literature and continue to evolve (e.g. modern frameworks like VGG-NT drop PnP-RANSAC and rendering-based refinement in favour of direct network-based pose prediction). Our geometrically consistent splats are compatible with these frameworks and could improve them. We use PnP-RANSAC as a proxy for measuring alignment error between two sets of *raw* splats predicted by any pixel-aligned method. Other heuristics from surfel-registration literature could yield faster or more accurate inference but are out of this paper’s scope.
>
> **Outdoor scene and object-centric reconstructions**
>
> The reviewer in the summary suggests that our \$L\_{orient}\$ encourages orientation of the neighbouring Gaussians to be similar. We want to clarify that this is not correct and might have led the reviewer to assume that higher curvature poses a problem for our regularisation. Please note that the L_orient only says that the means of the neighbouring Gaussians should provide appropriate local surface normals. These surface normals should align with the Gaussian orientations that the network predicts. Curvatures are neither estimated not penalised. It can be seen that true surface discontinuities are captured in the estimated normals by our approach (see Figures 1,2 of the main paper and supplementary material).  Additionally, both Re10K and ScanNet have some objects (and humans) with reasonable curvature. Our approach estimates these curvatures correctly too. We will include some of these visualisations in the paper alongside CO3D results for all baselines upon acceptance.

---

> > ### Comment · Reviewer_S4Ch · 2025-08-05
> >
> > I'd like to thank the authors for the detailed response, and in particular for correcting my understanding of the orientation loss and highlighting the stronger empirical gains for pose estimation. In light of these, I would be willing to increase my rating slightly but would still not argue for acceptance given the concerns about technical contributions. In particular, the response in the rebuttal  putting this work in the same equivalence class as prior generalizable splatting methods and interpreting the review as being applied to all of these slightly misses the point. I feel “adding regularizations to NoPoSplat” is an accurate description of this work. and this is less impressive than, say Splat3r or NoPoSplat which helped remove assumptions about the annotations available at inference compared to prior works at the time. Of course, this is not to say that in general,  contributions such as “adding regularizations” cannot be significant, but the concern here is that these regularizations, although not explored for generalizable splatting, are common in a slightly broader context context as the other reviews also highlight.

---

> > > ### Author Response · Authors · 2025-08-07
> > >
> > > We thank the reviewer for carefully going through our response and are please to see that reviewer understands the regularisation and pose estimation improvement better. We want to highlight that estimated scene structure by our approach is also a drastic improvement on the prior art.
> > >
> > > We want to clarify the intent about **equivalence of contributions with prior work** which might be misinterpreted. We were aiming to highlight two things in our original response:
> > > 1. Our work identifies and fix degeneracies observed in **all** self-supervised 3D splat estimators. These degeneracies are not NoPoSplat specific but are also observed in Pixel-Splat, MV-Splat (as shown in figure 2), Splat3R, FLARE, Depth-Splat and Any-Splat (visualisation for additional frameworks will be added to supplementary material). The common culprit in all cases is the ill posed learning objective that needs to be fixed. Proposed regularisers are (at times trivially) adoptable to any modern splat generation framework and hence have broader applicability. They are not NopoSplat specefic. Ours framework is NoPoSplat (+2DGS) + regularisers the same way NoPoSplat is replacing the Unimatch backbone of MVSplat's with Dust3R. Important consideration here is broader implication of the seemingly obvious A+B type technical change in both cases.
> > > 2. Response also elaborated that relying on ill posed loss for learning,  existing frameworks  ** failed to estimate 2/3rds of the Gaussian parameters (orientations and scales) ** they set out to do.Our work argued that over-perameterisation requires regularisation in the context of the problem. It is a rather obvious statement in hindsight but it is being overlooked by the community in last two years.  Ours was the first approach that predicted scales and orientation of the 3D splats with some efficacy.
> > >
> > > We hope we have clarified the broader applicability of our work beyond NoPoSplat and showcased substantial boost attained in terms of both pose and depth estimation that our work achieved as a system.
> > >
> > > The reviewer is still left with the concern about technical novelty. We readily agree that regularisation proposed in this work are used in broad context for decades -- but never used for making generalisation splatting well posed and essential. We leave the reviewers and ACs to judge the requirements on mathematical novelty expected at NeurIPS over appreciating well designed, simple learning frameworks that challenge and rectify important design oversights in solving an important problems.
> > >
> > > We again thank the reviewer and ACs for their careful consideration and time and are happy to clarify any other concerns reviewer have which discussion phase is open.

---

### Decision · Program_Chairs · 2025-09-17

**Decision:**

Reject

**Comment:**

This paper considers the task of generalizable Gaussian splatting: predicting a Gaussian mixture from two unposed images in a feedforward manner. In particular, it converts the NoPoSplat approach to use 2D Gaussian surface elements and two regularization terms that self-supervise the normals (the predicted normals should be consistent with the piecewise planar normals in a local neighborhood) and self-supervise the means (the predicted means should lie on the pixel ray). The AC notes that these terms could be applied to other feedforward splatting approaches beyond NoPoSplat. The identified strengths include that the design choices are reasonable and well-motivated, providing geometric cues in addition to the existing photometric cues, that the experiments were extensive, and the ablation study persuasive. The identified weaknesses predominantly relate to the perceived incremental nature of the contribution, with some additional concerns about the experiments and results.

The initial ratings were negative (2x reject and 2x borderline reject). Upon reading all reviews, author responses, and engaging in considerable discussion with the authors, the reviewers maintained their consensus to reject the paper, albeit with one reviewer raising their score to borderline reject following clarifications from the authors. All reviewers concur that the paper is relatively incremental on account of its contribution being two regularizations that are common in adjacent domains to the studied task, and therefore does not reach the bar for publication at NeurIPS. The AC, on-the-whole, agrees with this assessment and sees no reason to override the consensus of the reviewers. In particular, the AC agrees with reviewers S4Ch, ugeC and NUzX that the proposed elements appear in a largely similar form in other very similar domains. For example, the use of Gaussian surface elements has become increasingly common since the 2DGS paper in the domain of 3D reconstruction and editing, and the normal orientation loss is a variation of normal self-supervision used in 2DGS and more broadly in works such as RefNeRF, where predicted normals are supervised by the gradient of the density field. These ideas are now part of the researcher's toolkit and applications thereof cannot be considered major contributions. Indeed, when originally presented, these techniques were part of more broadly innovative methods, rather than standalone.